# First estimates of inequality benchmark incomes for a range of countries

**Laurence S. J. Roope** [ID] *

University of Oxford, Oxford, United Kingdom

* laurence.roope@dph.ox.ac.uk

## Abstract

It is known that virtually all inequality measures imply the existence of a 'benchmark income', above which adding incremental income increases inequality, and below which it decreases inequality. Benchmark incomes can be interpreted as social reference levels that identify the richest individual for whom it would be just to subsidize their income. Despite the intuitive appeal of benchmark incomes, there have been hardly any empirical applications to date. This paper provides the first estimates of benchmark incomes for a range of contrasting countries and different inequality measures. All benchmark incomes lie far above official national poverty lines. The results suggest that economic growth together with falling inequality need not necessarily be poverty reducing.

**Data Availability Statement:** All relevant data are within the paper and its Supporting information files.

**Funding:** The author is supported by the Oxford NIHR Biomedical Research Centre, Oxford.

## 1. Introduction

A number of theoretical studies have considered how incremental increases in income, at specific points in the income distribution, affect inequality. Virtually all inequality measures (all that embody social preferences that satisfy a strong version of the Pigou-Dalton transfer property) are associated with a benchmark income or position, above which adding increments of income increases inequality, and below which it decreases inequality [1].

The benchmark income, guaranteed by the strong Pigou-Dalton transfer property, is a distinguishing feature of inequality measures that sets them apart from both poverty measures and social welfare functions [1]. Poverty measures and social welfare functions typically satisfy a weak monotonicity property, where adding an increment to an individual's income can never increase poverty or reduce social welfare [1]. In contrast, adding increments of income to those above benchmark incomes actually increases inequality, capturing the intuition that there is something socially undesirable about these increases.

Poverty lines are often criticised for being largely arbitrary, with little theoretical basis for choosing a particular poverty line. This applies to relative poverty lines based on, for example, some percentage of the median income, as used in many countries. It also applies to absolute poverty lines, such as those based on minimum baskets of goods, where a variety of judgements must be made, such as what sorts of goods to include in a basket, and how to make these comparable across countries and over time. An attractive feature of the benchmark income approach is that, for a given income distribution and inequality measure, the

**Competing interests:** The author has declared that no competing interests exist.

benchmark income arises naturally and is determined by the social preferences (including the strong transfer principle) embodied in the inequality measure. Thus, if one has chosen to use a particular inequality measure, and implicitly adopted its normative underpinnings, one must also accept the benchmark income that is implied by these underpinnings.

Explicit results on where in an income distribution benchmark incomes lie have been derived for a wide range of inequality measures [1–4]. Benchmark incomes can be interpreted as social reference levels for inequality, somewhat analogous to poverty lines, above which increases to incomes increase inequality, and below which they decrease inequality. Knowledge of the location of the benchmark income could be used, for example, to predict the impact on inequality of a subsidy to income at a particular point in the distribution. Benchmark incomes can be interpreted as signifying the richest person in society for whom it is just and fair to subsidize their income [4, 5]. In the case of relative inequality, this interpretation is based on subsidies financed via proportional taxation—which leaves relative inequality measures unchanged. In the case of absolute inequality, it is based on subsidies financed by a lump sum tax—which leaves absolute inequality measures unchanged. Thus, the benchmark income can be used to identify the richest person for whom it might be deemed fair to subsidize income financed by taxation. Equivalently, the benchmark income could be interpreted as signifying the poorest person whom it is just and fair *not* to subsidise their income.

Despite the intuitive appeal of benchmark incomes, there have been hardly any empirical applications to date. (A notable exception is Hoffmann (2001) [2], who estimated benchmark incomes for Brazil in 1999.) This paper therefore provides an empirical illustration of the benchmark income approach, conducted using data from the UNU-WIDER World Income Inequality Database (WIID). We employ a variety of inequality measures to estimate 2010 inequality levels, and corresponding benchmark incomes, for a wide range of specific countries.

## 2. Inequality measures and benchmark incomes

For a society of $n \geq 2$ individuals let $\mathbf{x} = (x_1, \cdots, x_n) \in \mathbb{R}^n_+$ denote the distribution of incomes. An inequality measure is a function that assigns to each income profile a nonnegative number, so that $I : \bigcup_{n \in \mathbb{N}} \mathbb{R}^n_+ \longrightarrow \mathbb{R}_+$. We denote the mean of income profile $\mathbf{x} \in \mathbb{R}^n_+$ by $\mu = \frac{1}{n} \cdot \sum_{i=1}^n x_i$, and the median income by $m$. Let $\varepsilon > 0$ denote an incremental increase in some individual $l$'s income.

We employ five inequality measures with contrasting normative properties. These measures, the benchmark incomes corresponding to them and some limiting values, are given in Table 1.

The measures include two 'relative' measures, $I_G(\cdot)$ and $I_{MLD}(\cdot)$, two 'absolute' measures, $I_{AG}(\cdot)$ and $I_V(\cdot)$; and a 'centrist' measure, $I_K(\cdot)$. 'Relative' inequality measures are those which are invariant under equiproportional increases in all incomes. By contrast, 'absolute' inequality measures are those which register no change when the same absolute amount of income is added to all incomes. 'Centrist' inequality measures (sometimes also referred to as 'intermediate' or 'compromise' measures) register an increase in inequality if all incomes increase equiproportionally, and a decrease if the same absolute amount of income is added to all incomes. In the context of a growing economy, 'relative' measures are widely deemed "rightest" and 'absolute' measures "leftist" [6].

## 3. Data and empirical methods

The WIID is arguably the most comprehensive and complete database of worldwide distributional data available [7]. Income decile share data were obtained from the WIID (version 3.0b)

**Table 1. Inequality measures and corresponding benchmark incomes.**

| | Formula | Benchmark Income |
|---|---|---|
| Gini coefficient | $I_G(\mathbf{x}) = 1 - \frac{1}{n}\left[\frac{\sum_{k=1}^{n} 2\left(n-k+\frac{1}{2}\right)x_k}{\sum_{i=1}^{n} x_i}\right]$ | $c_{\mathbf{x}} = \frac{\sum_{k=1}^{n} kx_k}{\sum_{i=1}^{n} x_i}$ |
| | | $\lim n \to \infty \ \frac{c_{\mathbf{x}}}{n} = \frac{1}{2}\left(I_G(\mathbf{x}) + 1\right)$ |
| Mean log deviation | $I_{MLD}(\mathbf{x}) = \frac{1}{n}\sum_{i=1}^{n}\ln\left(\frac{\mu}{x_i}\right)$ | $c_{\mathbf{x},\varepsilon} = \frac{\varepsilon}{\left(1+\frac{\varepsilon}{n\mu}\right)^n - 1}$ |
| | | $\lim \varepsilon \to 0 \ c_{\mathbf{x},\varepsilon} = \mu$ |
| Absolute Gini | $I_{AG}(\mathbf{x}) = \mu \cdot I_G(\mathbf{x})$ | $c_{\mathbf{x}} = m$ |
| Variance | $I_V(\mathbf{x}) = \frac{1}{n}\sum_{i=1}^{n}(x_i - \mu)^2$ | $c_{\mathbf{x},\varepsilon} = \mu + \frac{1}{2}\left(\frac{n-1}{n}\right)\varepsilon$ |
| | | $\lim \varepsilon \to 0 \ c_{\mathbf{x},\varepsilon} = \mu$ |
| Krtscha | $I_K(\mathbf{x}) = \frac{1}{n\mu}\sum_{i=1}^{n}(x_i - \mu)^2.$ | $c_{\mathbf{x},\varepsilon} = \mu + \frac{\sigma_{\mathbf{x}}^2}{2\mu} - \frac{\varepsilon(n-1)}{2n}$ |
| | | $\lim \varepsilon \to 0 \ c_{\mathbf{x},\varepsilon} = \mu + \frac{\sigma_{\mathbf{x}}^2}{2\mu}$ |

NOTE: These results have been proven in previous studies [1–4].

for 2010, which at the time of analysis was the most recent year with available data for most countries. These data were used to estimate inequality measures and their associated benchmark incomes for a selection of countries.

The aim of the study was to provide an empirical illustration of the benchmark income approach, for a selection of countries and using a range of inequality measures. As there are hardly any estimates of benchmark incomes in the literature, the aim was to include a diverse selection of countries that differed in terms of level of development, geographic region and inequality levels. Thus, for high-income countries, several countries were included that follow what might be termed an 'Anglo-Saxon' model, with a small welfare state and comparatively high inequality levels; as well as several examples of a 'Nordic' model with a larger welfare state and comparatively low inequality levels. As a contrast to both Anglo-Saxon and Nordic countries, several 'BRICS' countries were included, which are notable due to both their increasing role in the global economy and their often high levels of inequality.

Where possible, data on individual income shares, rather than household income shares, were used. This was the case for all countries apart from India and Russia. The data for India were based on household income, while it was unclear in the dataset whether the data for Russia were based on household or individual income shares.

The inequality and corresponding benchmark income estimates were performed by creating a synthetic income distribution for each country, using a smoothing algorithm within deciles developed by Shorrocks and Wan (2009) [8]. Using data from the World Bank Databank, the synthetic distribution for each country was then scaled up by GDP per capita in 2005 US$ at purchasing power parity. This approach is widely regarded as providing better estimates than the simple approach of assuming that all individuals within the same decile have the same income, which biases inequality estimates downwards [9, 10]. Inequality levels and corresponding benchmark incomes were then estimated for each country. As a sensitivity analysis, estimates were also made under the simple assumption that all individuals within the same decile have the same income. Note that, with regard to estimating the percentiles in which benchmark incomes lie, an additional limitation of this assumption is that, for most of the inequality measures, it makes it impossible to locate where benchmark incomes lie within a particular decile. Instead, estimates will typically lie between two deciles and it will not be possible to identify, for example, whether they lie in, say, the 62nd, 65th or 68th percentile. The one exception is the Gini coefficient; benchmark incomes associated with the Gini may still be

estimated in the interior of a decile. This is due to the relationship (see formula in Table 1) linking the Gini to the position of the benchmark income in the income distribution.

## 4. Results

The inequality estimates in Table 2 indicate substantial differences in inequality levels between countries, and in how the different inequality measures rank the countries. The 'relative' inequality measures broadly agree that the Nordic countries are the most equal and the BRICS the most unequal. The 'absolute' measures agree instead that India is the most equal country, and the USA the most unequal. The 'centrist' Krtscha measure agrees with the 'relative' measures that South Africa is the most unequal country but, like the 'absolute' measures, deems India the most equal country. All measures judge the USA as more unequal than any of the countries in the sample outside of the BRICS. Analogous, qualitatively similar, results under the assumption of equal incomes within deciles are provided in S1 Table.

Unlike relative inequality measures, absolute measures such as the Absolute Gini and the Variance are sensitive to the absolute gaps between incomes. This means that in high-income countries absolute inequality is generally very high, even if relative inequality is very low, as the absolute gaps between incomes will generally still be high [10]. Incomes in South Africa are much lower than in the USA. Thus, even though the relative gaps between incomes are extremely high in South Africa (much higher than in the USA), the absolute size of the gaps between incomes is much lower than in the USA.

Turning to our main focus, the benchmark incomes implied by the Gini coefficient lie within the 62nd–85th percentile. The benchmark incomes implied by the Variance range from the 55th–78th percentile. Consistent with Table 1, these percentiles conform exactly with those corresponding to the MLD. The benchmark incomes implied by Krtscha's measure lie in the 66th-94th percentile. As shown in Table 1, for large $n$, the benchmark income percentiles

**Table 2. Inequality and benchmark percentiles in 2010.**

| | Inequality | | | | | Benchmark percentiles | | | | | Official poverty line percentile |
|---|---|---|---|---|---|---|---|---|---|---|---|
| | $I_G$ | $I_{MLD}$ | $I_{AG}$ | $I_V$ | $I_K$ | $p_G$ | $p_{MLD}$ | $p_{AG}$ | $p_V$ | $p_K$ | |
| *Nordic* | | | | | | | | | | | |
| Norway | 0.235 | 0.097 | 10,982 | 452.355 | 9,671 | 61.7 | 58.4 | 50 | 58.4 | 68.3 | 10.5 |
| Sweden | 0.241 | 0.101 | 8,211 | 241.151 | 7,067 | 62.0 | 56.4 | 50 | 56.4 | 66.6 | 15.4 |
| Denmark | 0.268 | 0.160 | 8,664 | 276.287 | 8,533 | 63.4 | 54.9 | 50 | 54.9 | 67.0 | 12.1 |
| *Anglo-Saxon* | | | | | | | | | | | |
| UK | 0.328 | 0.183 | 10,751 | 496.620 | 15,147 | 66.4 | 62.5 | 50 | 62.5 | 75.7 | 17.1 |
| Ireland | 0.332 | 0.186 | 12,195 | 649.859 | 17,666 | 66.6 | 62.9 | 50 | 62.9 | 77.6 | 15.2 |
| USA | 0.409 | 0.315 | 17,974 | 1445.318 | 32,886 | 70.5 | 63.2 | 50 | 63.2 | 78.7 | 15.1 |
| *BRICS* | | | | | | | | | | | |
| Russia | 0.397 | 0.260 | 5,652 | 153.472 | 10,786 | 69.9 | 68.0 | 50 | 68.0 | 80.1 | 12.5 |
| India | 0.417 | 0.287 | 1,283 | 9.607 | 3126 | 70.9 | 70.2 | 50 | 70.2 | 86.2 | 25.6 |
| Brazil | 0.536 | 0.525 | 5,405 | 207.683 | 20,577 | 76.8 | 72.5 | 50 | 72.5 | 89.8 | 16.1 |
| S. Africa | 0.696 | 0.990 | 6,628 | 505.560 | 53,125 | 84.8 | 78.0 | 50 | 78.0 | 94.7 | 53.2 |

NOTES: 1. Inequality estimates and benchmark percentiles based on WIID / Author's calculations; 2. Official poverty line percentiles are equivalent to official national poverty headcount ratios. For most countries, these estimates were obtained from the World Bank [11] for 2010. For India, the estimate is the average of estimates for 2009 and 2011. UK estimate is from the Office for National Statistics [12]. US estimate is from the US Census Bureau [13]. Brazil does not have an official national poverty line; estimate is from [14] based on an average of regional poverty lines; 3. $I_V$ is expressed in millions; 4. Benchmark percentiles $p_{MLD}$, $p_V$ and $p_K$ are based on the limits of the corresponding benchmark incomes as $\varepsilon \to 0$.

**Table 3. Correlations between inequality measures and benchmark percentiles.**

|  | $I_G$ | $I_{MLD}$ | $I_{AG}$ | $I_V$ | $I_K$ | $p_G$ | $p_{MLD}$ | $p_{AG}$ | $p_V$ | $p_K$ |
|---|---|---|---|---|---|---|---|---|---|---|
| $I_G$ | 1 | 0.968*** | -0.301 | 0.046 | 0.789*** | 1 | 0.944*** | 0 | 0.944*** | 0.950*** |
| $I_{MLD}$ |  | 1 | -0.255 | 0.056 | 0.855*** | 0.968*** | 0.860*** | 0 | 0.860*** | 0.855*** |
| $I_{AG}$ |  |  | 1 | 0.910*** | 0.274 | -0.301 | -0.456 | 0 | -0.456 | -0.396 |
| $I_V$ |  |  |  | 1 | 0.527 | 0.046 | -0.115 | 0 | -0.115 | -0.039 |
| $I_K$ |  |  |  |  | 1 | 0.788*** | 0.603* | 0 | 0.603* | 0.620* |
| $p_G$ |  |  |  |  |  | 1 | 0.944*** | 0 | 0.944*** | 0.950*** |
| $p_{MLD}$ |  |  |  |  |  |  | 1 | 0 | 1 | 0.986*** |
| $p_{AG}$ |  |  |  |  |  |  |  | 1 | 0 | 0 |
| $p_V$ |  |  |  |  |  |  |  |  | 1 | 0.986*** |
| $p_K$ |  |  |  |  |  |  |  |  |  | 1 |

NOTES: 1. Source: WIID / Author's calculations; 2. *, ** and *** indicate statistical significance at the 10%, 5% and 1% levels, respectively; 3. "1" and "0" indicate exact relationship, i.e., respectively, perfect and zero correlation.

implied by the Gini are a linear function of $I_G(\cdot)$. This relationship, and the lack of such one-to-one correspondence for the other measures, is apparent in Table 2. Analogous results under the assumption of equal incomes within deciles are provided in S1 Table.

To further explore the relationships between the measures and their implied benchmark percentiles, correlation coefficients between each of the measures and their benchmark percentiles are reported in Table 3.

As expected, the Gini coefficient is the only measure that is perfectly correlated with its benchmark income percentile. This means that the Gini coefficient's benchmark percentiles are consistent with the measure itself, in the sense that higher inequality necessarily means that subsidies can take place in higher percentiles before they become disequalizing—arguably an attractive property for an inequality measure. There is, however, a strong and highly statistically significant positive correlation (0.860) between the MLD and its benchmark percentile, and a moderately strong positive and borderline statistically significant correlation (0.620) between the Krtscha and its benchmark percentile. There is no correlation between the Absolute Gini and its benchmark percentile, which necessarily contains the median income, and we find no empirical evidence of any statistically significant correlation between the Variance and its benchmark percentile.

Strikingly, it is apparent from both Table 2 and, especially, from Table 3, that while the various measures rank countries quite differently with respect to inequality, ordering the countries according to the measures' benchmark percentiles provides very similar rankings. The point estimates of the pairwise correlation coefficients for the benchmark percentiles corresponding to each of the Gini, MLD, Variance and Krtscha with one another, are all >0.94 and statistically significant at the 1% level. There is no correlation, of course, between the benchmark percentiles implied by the Absolute Gini and those implied by any other measures, since the Absolute Gini's benchmark percentile is constant.

It is also apparent from Tables 2 and 3 that ordering countries according to the size of their benchmark percentiles, as implied by any of the Gini coefficient, MLD, Variance or Krtscha, provides very similar rankings to the Gini coefficient itself. As we have already seen, the rankings implied by the Gini coefficient's benchmark percentiles are identical to those of the Gini itself. The pairwise correlation coefficients between the Gini coefficient and the critical percentiles implied by the MLD, Variance and Krtscha are, respectively, 0.944, 0.944 and 0.950, all significant at the 1% level.

Thus, apart from the Absolute Gini, the measures broadly agree that the lower the 'relative' inequality is according to the Gini coefficient, the further down the income distribution subsidies to income must be in order for inequality to decrease. Where the measures do not agree, is in quite how far down the income distribution this point must be.

Apart from the Absolute Gini, the measures broadly agree that benchmark percentiles are generally highest in the BRICS (the countries with the highest 'relative' inequality) and lowest in the Nordic countries (the countries with the lowest 'relative' inequality). 'Relative' inequality in South Africa is found to be so high that, all else equal, even increasing incomes in the 77th, 77th, 84th and 94th percentiles would reduce inequality according to, respectively, the MLD, Variance, Gini coefficient and Krtscha. In contrast, in Sweden, increasing incomes in the 57th, 57th, 62nd and 67th percentiles would increase these respective inequality measures.

By way of comparison, Table 2 also contains estimates from the World Bank [11] and other sources ([12] for UK estimates; [13] for US estimates and [14] for Brazil estimates) of the poverty headcount ratio in 2010, based on the official poverty line—which is equivalent to the percentile of the income distribution in which the official poverty line lay. The benchmark incomes in all countries generally lay far above the official poverty line. The only exception is that the poverty line in South Africa, in the 53rd percentile, was slightly above the Absolute Gini's 50th percentile benchmark—but well below the benchmark percentiles implied by all the other inequality measures. For example, across the ten countries, the benchmark percentiles implied by the Gini coefficient lay on average 50 percentiles above the official poverty line percentile (ranging from 32 to 61 percentiles).

## 5. Conclusions

This is one of the first studies to illustrate where benchmark incomes lie in practice, across a selection of contrasting countries. All benchmark incomes for all countries lay far above official poverty lines. Across the ten countries studied, on average, half of the income distribution lay above the official poverty line but below the benchmark income implied by the Gini coefficient. The interpretation of the benchmark income as signifying the richest person for whom it might be fair to subsidize income has potential for informing redistributive policies.

Arguably, though economic growth is never confined to a single individual or percentile, benchmark percentiles may also be suggestive of the likely impact on inequality of certain growth-promoting policies. For example, in many developing countries, programmes to improve the quality of education, or infrastructure, in lagging rural areas might be expected to promote growth predominantly in parts of the income distribution below any of the benchmark percentiles derived in this paper. Such growth programmes would therefore be expected to reduce inequality. Similarly, the World Bank's explicit prioritisation in its 'shared prosperity' agenda of growing the incomes of the poorest 40% must be expected to reduce inequality according to a wide range of inequality measures [15].

An important implication of this study, however, is that economic growth alongside falling inequality need not necessarily be poverty reducing. If, due to a particular pattern of economic growth, gains are made mainly to incomes above the poverty line but below the benchmark income, inequality will fall but poverty will not. As the results in this study emphasise, incomes above the poverty line but below benchmark incomes typically constitute a large proportion of a country's income distribution. Growing these incomes can reduce inequality, even of the absolute kind, but not poverty. Thus, policy-makers should be careful not to assume that an increase in mean incomes, together with falling inequality (according to any of a wide range of measures), will necessarily result in people being lifted out of poverty. Instead, it is important to examine the overall pattern of growth, and in which percentiles of the economy it occurs.

## Supporting information

**S1 Table. Inequality and benchmark percentiles in 2010 under assumption of equal incomes within deciles.**
(DOCX)

**S1 Data.**
(XLSX)

## Acknowledgments

My thinking on this topic has benefited from conversations with Sabina Alkire, Natalie Quinn, Andrew Berg, Indranil Dutta, John McHale, Jonathan Temple, Kaushik Basu, Shasi Nandeibam, and Horst Zank—I am indebted to each of them. The paper has also benefited from discussions with various participants at the CSAE Conference 2015: Economic Development in Africa, held at the University of Oxford during 22–24 March 2015, the 72nd European Meeting of the Econometric Society, held in Lisbon during 21–25 August 2017, the Recent Developments in Distributional Analysis Workshop, held at the University of Leeds in April 2019, and at seminars at ETH Zürich, the University of Oxford and the National University of Ireland, Galway.

## Author Contributions

**Conceptualization:** Laurence S. J. Roope.

**Formal analysis:** Laurence S. J. Roope.

**Investigation:** Laurence S. J. Roope.

**Methodology:** Laurence S. J. Roope.

**Project administration:** Laurence S. J. Roope.

**Software:** Laurence S. J. Roope.

**Writing – original draft:** Laurence S. J. Roope.

**Writing – review & editing:** Laurence S. J. Roope.

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
