## [Decision Letter · Decision Letter 0]

30 Nov 2020

PONE-D-20-22441

First estimates of inequality benchmark incomes for a range of countries

PLOS ONE

Dear Dr. Roope,

Thank you for submitting your manuscript to PLOS ONE. After careful consideration, we feel that it has merit but does not fully meet PLOS ONE’s publication criteria as it currently stands. Therefore, we invite you to submit a revised version of the manuscript that addresses the points raised during the review process.

In addition to the referee report provided below, I have consulted another colleague for his/her advice.  The colleague provided the following suggestions:

1. The author may find this book useful / interesting: Deaton's *The Analysis of Household Surveys* (Ch. 3)

2. Some thoughts: - would benchmark incomes *also* signify the *poorest* person whom it is just and fair *to tax*? This might be interesting to a broader range of public economists and policymakers!

- inequality and social welfare are not (necessarily) the same thing. It would be good to explain what benchmark incomes have to do with various desirable properties of social welfare functions, or with the canonical social welfare functions.

- poverty lines come in for a lot of criticism that could also apply to benchmark incomes. The argument would be strengthened by explaining why benchmark incomes are preferable.

- is the author looking at household or individual incomes? Given differences in labor sharing within households across countries, the ranking of countries could depend on the reporting unit.

I would like you to address these concerns in the revised version of your paper.

We look forward to receiving your revised manuscript.

Kind regards,

Wing Suen

Academic Editor

PLOS ONE

Journal Requirements:

Reviewers' comments:

Reviewer's Responses to Questions

**Comments to the Author**

1. Is the manuscript technically sound, and do the data support the conclusions?

Reviewer #1: Yes

2. Has the statistical analysis been performed appropriately and rigorously? 

Reviewer #1: Yes

3. Have the authors made all data underlying the findings in their manuscript fully available?

Reviewer #1: No

4. Is the manuscript presented in an intelligible fashion and written in standard English?

Reviewer #1: Yes

5. Review Comments to the Author

Reviewer #1: The author adopted smoothing algorithm proposed by Shorrocks and Wan (2008) to compute various inequality measures and benchmark incomes for 10 countries and their correlations between inequality measures and benchmark incomes using WIID 2010 data. The findings suggest the benchmark incomes lie far above the poverty line and therefore imply that any policies enhancing incomes for those with incomes between these two numbers cannot reduce poverty.

The focus on the paper is more on the quantitative side rather than the policy side. I have the following suggestions regarding the data and discussion:

1. Can the author elaborate more about the choice of 2010 data and the 10 countries?

2. How much can we gain by using smoothing algorithm over constant-income-within-decile approach? Can the author provide more details about their differences or even generate the rankings using the latter approach?

3. Can the author provide the poverty line figures for the 10 countries parallel to the benchmark incomes? This can definitely show the policy-makers a clearer picture.

4. It would benefit the readers if the author can discuss more about the role of correlation between inequality measures (I) and benchmark percentiles (p) in Table 3. I understand if the I-I, I-p, or p-p pairs are highly correlated, then the corresponding rankings are similar. But which one a government should use? As the benchmark income for AG is a constant and thus uncorrelated with all I and p, does it mean that it is not reliable and thus the government should not use it?

5. Any explanation that USA’s wage inequality is the highest under AG and V measures? They triple S. Africa’s numbers but Gini’s measure shows the opposite. It would be better if the author can remind the readers when they use AG and V for analysis.

6. The conclusion part is a bit confused. It is said that “equalizing growth” cannot help reduce poverty, and that the incremental income between poverty line and benchmark incomes cannot help reduce poverty. What is the relationship between “equalizing growth” and “incremental income between poverty line and benchmark incomes”? Can you define “equalizing growth”? If a policy can increase income between poverty line and benchmark incomes, there is chance that the income below poverty line can also increase. It would be better if the author can cite some papers relating to “equalizing growth” policy that fails to ameliorate poverty.

There are some typos in the paper:

1. Line 71: I_(MLD) instead of I_T

2. Line 109: “Iv is expressed in million (square)”?

3. Line167: (57)th, (57)th instead of 56th, 56th as these are the thresholds for increasing inequality due to an increase in income, right?

6. PLOS authors have the option to publish the peer review history of their article (what does this mean?). If published, this will include your full peer review and any attached files.

Reviewer #1: No

---

## [Author Response · Author response to Decision Letter 0]

1 Feb 2021

See uploaded response to reviewers document.

---

## [Decision Letter · Decision Letter 1]

22 Feb 2021

First estimates of inequality benchmark incomes for a range of countries

PONE-D-20-22441R1

Dear Dr. Roope,

We’re pleased to inform you that your manuscript has been judged scientifically suitable for publication and will be formally accepted for publication once it meets all outstanding technical requirements.

Kind regards,

Wing Suen

Academic Editor

PLOS ONE

Additional Editor Comments (optional):

The paper was revised in a way that has adequately addressed the comments of the referees. I would recommend accepting this paper subject to correction of the minor typo pointed out in the referee's report.

Reviewers' comments:

Reviewer's Responses to Questions

**Comments to the Author**

1. If the authors have adequately addressed your comments raised in a previous round of review and you feel that this manuscript is now acceptable for publication, you may indicate that here to bypass the “Comments to the Author” section, enter your conflict of interest statement in the “Confidential to Editor” section, and submit your "Accept" recommendation.

Reviewer #1: All comments have been addressed

2. Is the manuscript technically sound, and do the data support the conclusions?

Reviewer #1: Yes

3. Has the statistical analysis been performed appropriately and rigorously? 

Reviewer #1: Yes

4. Have the authors made all data underlying the findings in their manuscript fully available?

Reviewer #1: Yes

5. Is the manuscript presented in an intelligible fashion and written in standard English?

Reviewer #1: Yes

6. Review Comments to the Author

Reviewer #1: The author has addressed all the comments by (1) adding a supplementary table showing the constant-income-within-decile approach, (2) adding poverty lines in Table 2, (3) clarifying the choice of data and (4) rewriting the conclusion about the confusion related to equalizing growth. I have no further comment except one typo in the note of the table on P9 Line181 - it is p_MLD instead of p_T.

7. PLOS authors have the option to publish the peer review history of their article (what does this mean?). If published, this will include your full peer review and any attached files.

Reviewer #1: No

---

## [Editor Report · Acceptance letter]

24 Feb 2021

PONE-D-20-22441R1 

First estimates of inequality benchmark incomes for a range of countries 

Dear Dr. Roope:

I'm pleased to inform you that your manuscript has been deemed suitable for publication in PLOS ONE. Congratulations! Your manuscript is now with our production department. 

Kind regards, 

on behalf of

Professor Wing Suen 

Academic Editor

PLOS ONE